# Structure and Dynamics of the Unassembled Nucleoprotein of Rabies Virus in Complex with Its Phosphoprotein Chaperone Module

**DOI:** 10.3390/v14122813

**Published:** 2022-12-16

**Authors:** Francine C. A. Gérard, Jean-Marie Bourhis, Caroline Mas, Anaïs Branchard, Duc Duy Vu, Sylvia Varhoshkova, Cédric Leyrat, Marc Jamin

**Affiliations:** 1Institut de Biologie Structurale (IBS), Université Grenoble Alpes, CEA, CNRS, 71 Avenue des Martyrs, 38000 Grenoble, France; 2Integrated Structural Biology Grenoble (ISBG), Université Grenoble Alpes, CNRS, CEA, EMBL, 71 Avenue des Martyrs, 38000 Grenoble, France; 3Institut de Génomique Fonctionnelle, Université de Montpellier, CNRS, INSERM, 34094 Montpellier, France

**Keywords:** rabies virus, *Mononegavirales*, phosphoprotein, nucleocapsid assembly, X-ray crystallography, small-angle X-ray scattering, molecular dynamics simulation

## Abstract

As for all non-segmented negative RNA viruses, rabies virus has its genome packaged in a linear assembly of nucleoprotein (N), named nucleocapsid. The formation of new nucleocapsids during virus replication in cells requires the production of soluble N protein in complex with its phosphoprotein (P) chaperone. In this study, we reconstituted a soluble heterodimeric complex between an armless N protein of rabies virus (RABV), lacking its N-terminal subdomain (N_NT-ARM_), and a peptide encompassing the N^0^ chaperon module of the P protein. We showed that the chaperone module undergoes a disordered−order transition when it assembles with N^0^ and measured an affinity in the low nanomolar range using a competition assay. We solved the crystal structure of the complex at a resolution of 2.3 Å, unveiling the details of the conserved interfaces. MD simulations showed that both the chaperon module of P and RNA-mediated polymerization reduced the ability of the RNA binding cavity to open and close. Finally, by reconstituting a complex with full-length P protein, we demonstrated that each P dimer could independently chaperon two N^0^ molecules.

## 1. Introduction

Rabies is a zoonotic, incurable brain disease that, despite the availability of an efficient vaccine and post-exposure prophylaxis, continues to kill tens of thousands of people every year according to WHO [1]. Its etiological agent, the eponymous rabies virus (RABV), is a prototypic member of the *Rhabdoviridae*, a large family of nonsegmented negative-sense RNA viruses that includes vesicular stomatitis virus (VSV) [2], Chandipura virus [3], and several underexplored viruses infecting various animals and plants [4]. Because these viruses share a similar organization of their genome and of their virion as well as similar strategies for the transcription, replication, and encapsidation of their genome with other negative-sense RNA viruses, the family *Rhabdoviridae* is classified into the order *Mononegavirales* (*MNV*) with, inter alia, the families *Paramyxoviridae* (including measles virus, mumps virus, parainfluenza viruses, and the zoonotic Nipah and Hendra viruses), *Pneumoviridae* (including respiratory syncytial virus and metapneumovirus), and *Filoviridae* (including the zoonotic Ebola and Marburg viruses) [5,6]. The order *Mononegavirales* is now classified with all other known segmented and nonsegmented negative-sense RNA viruses in the phylum *Negarnaviricota* [6].

The RABV genome of 11.9 kb comprises five transcription units, which encode five structural proteins: the nucleoprotein (N), the phosphoprotein (P), the matrix protein (M), the surface glycoprotein (G), and the large RNA-dependent RNA polymerase (L-RdRP). The multiplication of the virus in appropriate host cells involves (i) the transcription of the genomic RNA to generate messenger RNAs, (ii) the replication of the genome through the intermediate production of positive-sense antigenomic RNAs, which are subsequently used as a template to generate new genomic RNAs, and (iii) the packaging of both genomic and antigenomic RNAs into linear homopolymers of N, forming helical and flexible ribonucleoprotein complexes, named nucleocapsids (NCs) [7,8]. A unique molecular machine involving three viral proteins (N-, P-, and L-RdRP) carries out these three processes within cytoplasmic membrane-less viral factories, called Negri’s bodies [9]. In particular, the assembly of novel NCs is an essential step in the infectious cycle of these viruses because only the genomic and antigenomic RNAs packaged in NC can serve as template for RNA synthesis by the polymerase complex (composed of P- and L-RdRP) [10,11,12]. Indeed, if L-RdRP catalyzes RNA synthesis, as well as the processing of mRNAs, including the synthesis and methylation of the 5′ cap and the synthesis of the 3′ poly-A tail, it is unable to proceed efficiently on naked RNAs and requires the presence of both P and N for processive transcription and replication [10]. Thus, genome replication requires the production of the unassembled RNA-free N protein, named N^0^, in the form a complex named the N^0^−P complex [13,14], in which P prevents the polymerization of N and the illegitimate assembly with cellular RNA until N is transferred to nascent viral RNA [15].

RABV P is an essential multi-functional protein with a modular architecture [16], which acts both as a cofactor of the L-RdRP and as a chaperone of N^0^, but also hijacks cellular components [17] and counteracts the host antiviral responses [18]. RABV P forms dimers [19]; each protomer consists of a long N-terminal intrinsically disordered region (P_NTR_, aa 1–90) and a C-terminal region (P_CTR_) consisting of two folded domains, the multimerization domain, P_MD_ (aa 91–131), and the C-terminal domain (aa 186–297), P_CTD_, which are connected by a flexible linker (aa 132–185) [20,21,22]. The P protein works as a hub, in which structural and functional modules interact independently with different partners; the extremity of P_NTR_ consists of the chaperon module (P_CM_—aa 1–40) that binds N^0^ [23], while most of the remaining part of P_NTR_ binds L-RdRP (residues 40–88) [24], whereas P_CTD_ binds to the polymeric N−RNA complexes (residues 185–297) [21]. The N-terminal segment of P_NTR_ is sufficient for keeping N in a monomeric soluble form [23,25,26,27,28], but it is also believed that the N^0^−P complex is recruited at the site of RNA synthesis by the attachment of P_CTD_ to the template [29]. Different modules or motifs of RABV P bind the dynein LC8 [17], the focal adhesion kinase [30], the ribosomal protein L9 [31], the mitochondrial complex I [32], IRF-3 [33], STAT1 [34], PML [33,35], nuclear import and export factors [36], BECN1 [37], and Cdc37/Hsp90 complex [38]. RABV N is composed of two globular domains that are connected by a hinge, creating an RNA binding groove [39]. It also comprises two subdomains, an N-terminal extension, named the N_NT-ARM_**,** and an internal loop in the C-terminal domain, named the N_CT-ARM_, that dock to neighboring subunits in the polymeric form and thereby stabilize the assembled NC [39].

Previously, working with VSV, we developed a strategy to reconstitute a core N^0^−P complex between a nucleoprotein truncated of its N-terminal arm (VSV N_NT-ARM_: aa 1–21), VSV N_∆21_, and a peptide of 60 residues (VSV P_60_) derived from P that comprised the chaperone module (P_CM_) and we solved the crystal structure of the complex [40]. The reconstituted VSV N_∆21_^0^−P_60_ complex was monomeric in solution, but assembled into RNA-free 10-mers circular complexes in the crystal, resembling the recombinant circular NC-like complexes [41]. In the crystal structure, P_CM_ extended in a groove of the C-terminal domain and formed a long α-helix bound at the interface of the N- and C-terminal domains of N. The structure unveiled a mechanism for preventing the assembly of N by interfering with the binding of the N_NT-ARM_ from the adjacent N_i-1_ protomer and with the binding of N_CT-ARM_ from the N_i+1_ protomer [40]. The protrusion of the C-terminal part in the RNA binding groove of the N protein suggested a direct interference with RNA binding [40]. We also showed that no additional interaction was detected between full-length P and N^0^, that the phosphorylation of P_NTR_ by casein kinases (Ser60, Thr62, and Ser64) had no effect on the interaction with N^0^ and that dimeric P could bind and chaperone one or two N^0^ molecules depending of the concentration of N^0^ in the assembly conditions [42]. The structure of the N^0^−P complex has since then been obtained for several other RNA viruses in the families *Paramyxoviridae* [43,44,45], *Pneumoviridae* [46], and *Filoviridae* [26,28,47], showing that the interference of P_CM_ with the N_NT-ARM_ and N_CT-ARM_ binding is a conserved feature within the order *Mononegavirales*, although the structure and position of P_CM_ on the surface of N varied between the viruses. In VSV crystal structure, the N^0^ protein was in a close conformation, unable in this state to accommodate the incoming RNA molecules, whereas, by contrast, in the N^0^−P complex of these other viruses, the N^0^ molecule was trapped in an open conformation, unable to strongly bind RNA, but ready to grasp an incoming RNA molecule.

The comparison of the structure of the VSV N_∆21_^0^−P_60_ complex [40] with that of the VSV N−RNA complex [41] and with those of the N^0^−P complexes of other viruses [26,28,43,44,45,46,47] raised several questions. First, despite the binding of P_CM_ and the truncation of the N_NT-ARM_, the VSV N_∆21_ protein assembled in a polymeric form in the crystal, suggesting that P_60_ was not sufficient to maintain the N^0^ protein in its monomeric state. Second, in the N_∆21_^0^−P_60_ crystal structure, the N protein was in its closed conformation, which was identical to its RNA-bound conformation, whereas in the structure of the other N^0^−P complexes, the N protein was in an open conformation [26,28,43,44,45,46,47]. This suggested the occurrence of a hinge motion in N allowing the insertion and release of the RNA molecule, but it raised the possibility that the assembly of VSV N into circular complexes had induced the closure of the protein. Third, in the VSV N_∆21_^0^−P_60_ complex, the N-terminal part of P_CM_ was bound in same groove of N_CTD_ as the N_NT-ARM_ and in the same orientation. In other known N^0^−P structures, the chaperone module (P_CM_) adopted an orientation opposite to that of the N-terminal arm of N (N_NT-ARM_), against leaving a doubt that the assembly of N could influenced the interaction with P_CM_ [26,28,43,44,45,46,47].

To address these questions, we studied the structure and the dynamics of the RABV N^0^−P complex. We used the same strategy for reconstituting the N^0^−P core complex than for VSV and NiV [40,43] that involved the independent expression and purification of truncated N and of a P fragment, their mixing, and the purification of the reconstituted complex. We hypothesized that a slightly longer disordered region at the C-terminus of the P peptide might prevent the assembly of N^0^ into rings by further masking the surface of N with the flexible part of the P peptide. We characterized the structure of the complex in a solution by SEC-MALLS and SEC-SAXS and solved the crystal structure. We implemented a fluorescence anisotropy assay to monitor the binding kinetics and equilibrium of the peptide. To better understand the hinge motion mechanism in the N protein and how the binding of P or the assembly into a polymeric N−RNA complex affected this conformational change, we performed molecular dynamics simulations with the different forms of N.

## 2. Materials and Methods

### 2.1. Bioinformatics

The amino acid sequences of Lyssavirus phosphoproteins were retrieved from the Uniprot database [48] and multiple sequence alignments (MSA) were performed with Clustal Omega using default parameters [49]. The calculation of the D-score, which provided a consensus prediction of 13 disorder algorithms available through webservers and defined the boundaries of structured domains, was performed as previously described [20]. The interface between N_D23_^0^ and P_68_ was analyzed with PDBePISA [50], structural alignment with PDBeFOLD [51], and computational alanine scanning mutagenesis was performed with FoldX [52].

### 2.2. Constructs

The plasmid (pET22b(+)) containing the gene of full-length RABV P (CVS-11 strain) (UniProt P22363) fused to a C-terminal two-amino linker and 6xHis-tag for expression in bacteria was previously described [20]. The cDNA encoding residues from 1 to 68 of RABV phosphoprotein were cloned into the pET28a (Novagen, Darmstadt, Germany) vector using NcoI and XhoI restriction sites in a frame with a downstream 6xHis-tag and a two amino-acid linker (Glu-Leu). A synthetic cDNA (Geneart, Regensburg, Germany) encoding the first 42 residues of RABV phosphoprotein with a cysteine substitution at position 41 (G41C), an N-terminal 6xHis tag, a SUMO tag and a tobacco etch virus protease (TEV) cleavage site were cloned into the pET22b expression vector (Novagen) using the NcoI and XhoI restriction sites. Point mutations were introduced in this construct by site-directed mutagenesis using the QuickChange II protocol (Agilent, Santa Clara, CA, USA). The cDNA encoding residues from 24 to 450 of rabies virus (strain CVS-11) nucleoprotein (UniProt Q8JXF6) were cloned into the pETM-40 (EMBL) vector using NcoI and XhoI restriction sites in frame with the upstream MalE gene encoding the maltose binding protein (MBP) and a TEV cleavage site. All the plasmids were verified using standard dideoxy sequencing.

### 2.3. Protein Expression and Purification

All the constructs were transformed into *E. coli* BL21 (DE3). The cells were grown at 37 °C in Luria Bertoni medium containing ampicillin 100 µg/mL (for pET22b constructs) or kanamycin 50 µg/mL (for pET28a and pETM-40 constructs) until the optical density at 600 nm reached 0.6. The protein expression was then induced by the addition of 0.5 mM isopropyl β-D-1-thiogalactopyranoside (IPTG) and cells were further grown at 18 °C for 18 h. The cells were harvested by centrifugation and the pellet was suspended in buffer A (50 mM Tris-HCl buffer at pH 7.5 containing 300 mM NaCl, and 0.2 mM Tris(2-carboxyethyl)phosphine (TCEP)) and supplemented with an EDTA-free Complete Protease Inhibitor Cocktail (Roche, Bâle, Switzerland). The cells were disrupted by sonication and the crude extract was cleared by centrifugation at 35,000× *g* at 4 °C for 30 min. For proteins containing a 6xHis-tag (RABV P_68_, P_42_, and P_FL_), the supernatant was loaded onto an Ni−NTA Agarose (Qiagen, Hilden, Germany) column equilibrated in buffer A. The column was washed with ten column volumes of buffer B (50 mM Tris-HCl buffer at pH 7.5 containing 1 M NaCl, 15 mM imidazole, and 0.2 mM TCEP) and the protein was eluted with buffer C (50 mM Tris-HCl buffer at pH 7.5 containing 150 mM NaCl, 400 mM imidazole, and 0.2 mM TCEP) and dialyzed overnight against buffer D at 4 °C (50 mM Tris-HCl buffer at pH 7.5 containing 150 mM NaCl and 0.2 mM TCEP). For RABV P_42_ containing a TEV site, the overnight dialysis against buffer D was performed in the presence of the TEV protease at an approximate weight ratio of 100:2 (fusion protein/TEV). After concentration with Vivaspin 20 (3 kDa molecular weight cut-off (MWCO), Sartorius, Göttingen, Germany) or with Amicon (molecular mass cutoff, 10 kDa) (Millipore, Burlington, MA, USA) concentrators, the protein was loaded on a S75 Superdex column (Cytiva, Malborough, MA, USA) equilibrated in buffer D. For N_∆23_−MBP, the supernatant was loaded onto an amylose resin column (New England BioLabs, Ipswich, MA, USA) equilibrated in buffer A. The column was washed with ten column volumes of buffer B and the protein was eluted with buffer E (50 mM Tris-HCl pH 7.5 containing 150 mM NaCl, 0.2 mM TCEP, 50 mM arginine, 50 mM glutamate, and 250 mM maltose). The fractions containing the protein were pooled and concentrated to 3 mg/mL with Amicon concentrators (molecular mass cutoff, 30 kDa) (Millipore), the protein was loaded on a S200 Superdex column (Cytiva) equilibrated in buffer D at 4 °C. Each protein sample was analyzed by SEC-MALLS (Multi-Angle Static Light Scattering—Dawn Heleos II, Wyatt, Santa Barbara, CA, USA) and the concentration was measured by on-line refractometry (Optilab T-rex refractometer, Wyatt).

### 2.4. Peptide Labelling with a Fluorescent Dye

A solution of concentrated His_6_-SUMO-TEV-P_42-G41C_ protein in buffer D was incubated for 20 min with a 10-fold molar excess of TCEP and flushed for 1 min with nitrogen gas (N_2_). A 10-fold molar excess of 5-carboxyfluorescein (5-FAM) maleimide (Thermo Scientific, Waltham, MA, USA) dissolved in dimethylsulfoxyde (DMSO) was added dropwise and the mixture was incubated overnight in the dark. A 10-fold molar excess of reduced glutathione was added and the solution was loaded on a PD10 desalting column (Econo-Pac^®^ 10DG columns, Bio-Rad, Hercules, CA, USA) and eluted with buffer D. The protein was cleaved with the TEV protease at an approximate weight ratio of 100:2 (fusion protein/TEV) and overnight incubation at 4 °C to remove the tag. The labeling efficiency was assessed using liquid chromatography coupled to electrospray ionization mass spectrometry (ISBG facility, Grenoble). The labeled peptide P_1–42, G41C_-FAM was analyzed by SEC-MALLS (Multi-Angle Static Light Scattering—Dawn Heleos II, Wyatt) and the concentration was measured by on-line refractometry (Optilab T-rex refractometer, Wyatt).

### 2.5. Reconstitution of the N_∆23_^0^–P_68_, N_∆23_^0^–P_42-FAM_, and N_∆23_^0^–P_FL_ Complexes

The purified MBP-TEV-N_Δ23_ was mixed with an excess of purified peptide (P_68_–His_6_, P_42_–FAM) or of purified P_FL_ and the mixture was incubated for 2 h at 4 °C. The MBP-tag was cleaved with the TEV protease at an approximate weight ratio of 100:2 (fusion protein/TEV) and overnight incubation at 4 °C. The solution was concentrated and loaded onto a S200 Superdex (Cytiva) column coupled to a short amylose resin (NEB) column equilibrated in 50 mM HEPES buffer at pH 7 containing 150 mM NaCl, 50 mM arginine, 50 mM glutamate, and 0.2 mM TCEP to completely remove cleaved MBP.

### 2.6. Size Exclusion Chromatography and Multiangle Laser Light Scattering (SEC-MALLS)

The size exclusion chromatography (SEC) combined with on-line detection using multiangle laser light scattering (MALLS) and refractometry is a method for measuring the absolute molecular mass of a particle in a solution that is independent of its dimensions and shape [53]. The SEC was performed at 20 °C with a Superdex 200 or Superdex 75 column (Cytiva) equilibrated with a 20 mM Tris-HCl buffer containing 150 mM NaCl and 0.2 mM TCEP and a flow rate of 0.5 mL.min^−1^. The column was calibrated with globular standard proteins of known hydrodynamic radius (R_h_) [54]. MALLS detection was performed on-line with a DAWN-HELEOS II detector (Wyatt Technology, Santa Barbara, CA, USA) using a laser emitting at 690 nm and protein concentration was measured on-line using differential refractive index measurements with an Optilab T-rEX detector (Wyatt Technology) and a refractive index increment, dn/dc, of 0.185 mL.g^−1^. The weight-averaged molecular mass was calculated using the ASTRA software (Wyatt Technology).

### 2.7. Competition Anisotropy Binding Assay

In the kinetic assay, the N_∆23_^0^−P_42-FAM_ complex at 100 nM in buffer F was mixed with the unlabeled P_42-G41C_ at a final concentration of 15 µM and fluorescence anisotropy was measured with a spectrofluorimeter (Quantamaster QM4CW Photon Technology International, Birmingham, NJ, USA) with polarizers and thermostated at 20 °C. The excitation and emission wavelengths were set at 495 nm and 520 nm, respectively. In the equilibrium binding assay, the N_∆23_^0^−P_42-FAM_ complex (100 nM) was mixed with serially diluted unlabeled peptides in black 96-well plates. The samples were then incubated for 24 h at room temperature and fluorescence anisotropy was measured using a plate reader (Clariostar, BMG LABTECH, Ortenberg, Germany) at 20 °C using an excitation filter at 485 nm and an emission filter at 535 nm. All the experiments were carried out in triplicate. The curve fitting was performed with the software program Dynafit [55].

### 2.8. X-ray Crystallography

The crystallization screenings of the N_Δ23_−P_68_ complex were carried out at the High Throughput Crystallization Laboratory of the EMBL Grenoble Outstation (HTX Lab.). The crystals of the RABV N_Δ23_−P_68_ complex were obtained at 20 °C in (0.02 M sodium/potassium phosphate and 0.1 M Bis Tris propane at pH 6.5 containing 20% *w*/*v* PEG 3350). The diffraction data were collected on the ID23-1 beamline at the ESRF (Grenoble, France). The N_Δ23_−P_68_ complex crystallized in space group P2_1_2_1_2 with one complex per asymmetric unit. The single crystals were harvested from the drop, briefly soaked in the reservoir solution supplemented with 25% glycerol and flash frozen in liquid nitrogen at 100 K before data collection. The data were processed using the program XDS [56] and scaled with the program Scala from the ccp4 suite [57]. The structure was solved by molecular replacement with the program Phaser [58] using a protomer of N extracted from the N−RNA crystal structure (2GTT.pdb) [39] as a search model. The overall structure was refined to a resolution of 2.3 Å using Coot [59] and Refmac5 [60] and Buster [61]. The quality of the model was checked with PROCHECK [62]. The data collection and refinement statistics are summarized in Table 1.

We re-refined the circular RABV N_11_−RNA crystal structure (PDB ID 2GTT) in order to correct a frameshift error present in the original entry. The error was detected by comparing the structure of one N subunit from the N_11_−RNA crystal structure with the N structure from the newly solved N_Δ23_^0^−P_68_ crystal structure (8B8V.pdb) and by performing MD simulations of an N−RNA trimer extracted from the N_11_−RNA crystal structure (2GTT.pdb), which showed significant unfolding of the N-terminal domains within a few hundred ns of MD. The N_11_−RNA structure was re-refined using a combination of manual rebuilding in Coot [59] based on the new N_Δ23_^0^−P_68_ crystal structure, molecular dynamics flexible fitting (MDFF) in ISOLDE [63], and repetitive rounds of restrained refinement in PHENIX [64] and Autobuster [61]. The new coordinates have been deposited in the PDB under the code 8FFR.

### 2.9. Small Angle X-ray Scattering (SAXS) and SEC-SAXS Experiments

The small angle X-ray scattering (SAXS) experiments were performed on the BioSAXS beamline BM29 and former beamline ID14-3 at the European Synchrotron Radiation Facility (ESRF), Grenoble, France and on the SWING beamline at SOLEIL, Paris, France. For direct SAXS experiments, the scattering from the buffer alone was measured before and after each sample measurement and used for background subtraction. The 1D scattering profiles were generated and buffer subtraction was carried out by the automated data processing pipeline available at the different beamlines. For SEC-SAXS experiments, the samples were loaded onto a Superdex^TM^ 200 increase 5/150 GL equilibrated with 50 mM HEPES buffer at pH 7.5 containing 150 mM NaCl and 0.5 mM TCEP. The program Foxtrot [65] was used to integrate and subtract the SEC-SAXS frames. All the data were analyzed with the program PRIMUS from the ATSAS 3.0.0 package [66]. The radius of gyration was determined with the program PRIMUS according to the Guinier approximation at low q values and the molecular weights were estimated based on the invariant V_c_ and R_g_ values [67].

For isolated RABV P_68_, an ensemble of 10,000 conformers was generated using the software Flexible-Meccano [68] and sidechains were added using SCWRL4 [69]. For the N_∆23_^0^−P_68_ complex, an ensemble of 6639 models was built by extracting snapshots of explicit solvent MD trajectories (see Methods—MD simulations), using a time step of 1 ns. For the N_∆23_^0^−P_68_ complex, several ensembles of 10,000 models were built with the software programs RANCH [70], PD2 [71], and SCWRL4 [69]. The theoretical SAXS patterns were calculated with the program CRYSOL [72] and ensemble optimization fitting was performed with GAJOE [70]. The optimum selected ensemble size and relative weights of the models were determined automatically by GAJOE.

### 2.10. Molecular Dynamics (MD) Simulations

The classical explicit solvent MD simulations were used to study the conformational dynamics of the N protein in its free form (N^0^), in complex with P_68_ (N^0^−P_68_), and in its RNA-bound form (N−RNA). The N^0^−P_68_ MD trajectories were additionally used to provide an ensemble of models suitable for fitting the SAXS data using EOM. The N^0^ and N^0^−P_68_ models were based on the N_∆23_^0^−P_68_ crystal structure (missing residues in N^0^ and P_68_ were constructed as random chains) and the simulated N−RNA model corresponded to a trimer of RNA-bound N protein extracted from the revised version of the 11-mer N−RNA ring crystal structure. All 3 systems were simulated in GROMACS [73] using an amber99SBws forcefield [74], which was designed to reproduce the properties of intrinsically disordered proteins. At the beginning of each simulation, the protein was immersed in a box of TIP4P2005 water, with a minimum distance of 1.0 nm between protein atoms and the edges of the box. The genion tool was used to add 150 mM NaCl. The long-range electrostatics were treated with the particle-mesh Ewald summation. The bond lengths were constrained using the P-LINCS algorithm. The integration time step was 5 fs. The v-rescale thermostat and the Parrinello–Rahman barostat were used to maintain a temperature of 300 K and a pressure of 1 atm. Each system was energy minimized using 1000 steps of steepest descent and equilibrated for 500 ps with restrained protein heavy atoms prior to production simulations; 4 or 5 independent MD trajectories were calculated for each system, representing a total aggregated simulation time of 3.1 µs (N^0^), 6.6 µs (N^0^−P_1-68_), and 2.5 µs (N−RNA). RMSD, RMSF, and principal component analysis (PCA) were calculated using GROMACS routines.

## 3. Results

### 3.1. The N-Terminal Chaperone Module of RABV P Is Intrinsically Disordered

In a recent study, we have showed by nuclear magnetic resonance (NMR) spectroscopy and small-angle X-ray scattering (SAXS) that the entire N-terminal region of RABV P (from aa 1 to 90) is intrinsically disordered both in isolation and within the full-length dimeric P protein [17]. However, multiple sequence alignments revealed that the region encompassing residues from 1 to 60 was conserved within the genus Lyssavirus (Figure 1A) and our meta-analysis of disorder predicted that the first 55 residues were structured (Figure 1B). Together with the previous studies conducted with rabies virus and other viruses, these results suggested that the N-terminal end contained a molecular recognition element (MoRE) involved in chaperoning the unassembled, RNA-free nucleoprotein [23,25,40,43,75,76]. On the basis of this information, we generated two constructs that encompassed the N-terminal 42 or 68 residues of P; the former was fused to N-terminal cleavable 6-histidine and SUMO tags (His6-SUMO-P_42_), while the latter was fused to a C-terminal two-amino acid linker (LE) and a 6-histidine tag (P_68_−His_6_) (Figure 1C). We purified both peptides using affinity chromatography on Ni−NTA resin and size-exclusion chromatography (SEC) on a Superdex S75 column. Each peptide eluted from the SEC column in a single peak and its weight-average molecular mass (M_w_) determined from static light scattering and refractometry measurements (SEC-MALLS) was in agreement with the theoretical molecular masses of the respective monomers (Figure 1D). For RABV P_68_, we recorded SAXS curves for scattering vector (q) values ranging from 0.05 to 4.0 nm^−1^ over a concentration range from 3.3 to 11.0 mg.mL^−1^ (Figure 1E). The Guinier plots at low q values (q.R_g_ < 1.3) were linear (Figure 1F) and showed only a slight dependence on a protein concentration indicating the absence of aggregation but the presence of some interparticle interactions at the highest concentration (11 mg.mL^−1^) (Figure 1G). To optimize the signal-to-noise ratio and eliminate any possible structure factor contribution, we merged the curves at the different concentrations [77] (Figure 1H) and used the software GAJOE to select ensembles of a limited number of conformers from a large initial ensemble that reproduced the experimental SAXS data [70]. The initial ensemble of the all-atom models generated with Flexible-Meccano [68] displayed a Gaussian distribution of R_g_ and D_max_ values as expected for a random coil (Figure 1I,J). The selected ensembles of the 12 conformers adequately reproduced the SAXS curve (χ^2^ = 0.824) (Figure 1H), while the R_g_ and D_max_ distributions of the selected ensembles were comparable to those of the initial ensemble, confirming that P_68_ obeyed random coil statistics (Figure 1I,J). Figure 1K shows a representative ensemble of 12 conformers.

### 3.2. The Chaperone Module of RABV P Blocks N in a Monomeric Closed Conformation

We expressed a truncated form of RABV N lacking its N-terminal arm (N_NT-ARM_—aa 1–23) in fusion with a TEV cleavable N-terminal MBP tag (Figure 2A) (MBP-N_Δ23_). We purified the protein using affinity chromatography on an amylose resin. We then mixed the preparations of MBP-N_Δ23_ and purified P_68_−His_6_, cleaved the MBP tag with the TEV protease, and purified the N_Δ23_^0^−P_68_ complex using SEC on a Superdex S200 column. We showed by SEC-MALLS that the N_Δ23_^0^−P_68_ complex eluted as a single, symmetric peak from the Superdex S200 column (Figure 2B). The M_w_ value of 58 ± 3 kDa obtained from light scattering and refractometry measurements was in agreement with the theoretical molecular mass of 56.7 kDa calculated for the heterodimer (Figure 2B). The M_w_ was constant throughout the chromatographic peak with M_w_/M_n_ ratio (where M_n_ is the number-average molecular mass) of 1.001, indicating a monodisperse species.

The N_Δ23_^0^−P_68_ complex crystallized with a single molecule in the asymmetric unit and the crystals diffracted up to 2.3 Å resolution (Table 1). The structure was solved by molecular replacement using the structure of an N protomer extracted from the circular N−RNA complex (PDB code: 2GTT) [39]. The N_Δ23_^0^−P_68_ complex contained no RNA and an analysis with the webserver PDBePISA revealed no intermolecular interactions between neighboring heterodimers besides the crystal contacts (Complex Formation Significance Score (CSS) = 1.0 for the interaction between N_Δ23_^0^ and P_68_ and CSS = 0.0 for all other contacts), demonstrating the absence of N polymerization in the crystal (Appendix A). The N_Δ23_ molecules arranged themselves side by side in linear arrays in the crystal and their N-terminal domains contact each other (Appendix A), but without forming a tight complex as in NC (Appendix A). The N_Δ23_ exhibited the typical *Mononegavirales* nucleoprotein fold with two globular domains, N_NTD_ and N_CTD_, connected by a hinge (Figure 2C). In the absence of neighboring N subunits, the entire N_CT-ARM_ (aa 350–400) was not visible in the crystal structure and a 5-residue loop in N_NTD_ (aa 155–159) was also absent from the crystal structure, suggesting that it was flexible in the absence of RNA as expected, since this loop interacted with one of the bound nucleotides in the N−RNA complex [39]. Two parts of P_68_ were clearly visible in the electronic density; residues from 4 to 15 were bound in a groove of N_CTD_ with residues from 9 to 12 forming a helical turn, while residues from 20 to 39 formed a long α helix that docked at the interface between N_NTD_ and N_CTD_ (Figure 2C). The four residue-connecting loops (aa 16–19) were less well-defined in the electron density map, suggesting that it was slightly flexible in the crystal (Figure 2C). The complex was stabilized by multiple interchain interactions. The binding of the P_68_ N-terminal moiety involved three H-bonds and four salt-bridges and buried a surface area of 1190 Å^2^, whereas the binding of the P_68_ C-terminal moiety involved thirteen H-bonds and three salt bridges and buried a surface area of 1617 Å^2^ (Appendix A). The interface regions of N and some interface residues of P appeared to be conserved among the Lyssavirus (Figure 2D,E).

The structural alignments using PDBeFold showed that N_CTD_ in our refined N_Δ23_^0^−P_68_ structure was similar to that in the N−RNA complex [39] (RMSD = 0.84 Å) (Figure 2F), while the N_NTD_ exhibited some differences with residues from 35 to 103 and from 189 to 200 being out-of-register in the structural alignment despite an RMSD of 0.95 Å (Figure 2G). An analysis of the original electron density map of the N−RNA complex [39] revealed that the absence of density for residues from 105 to 118 and 185 to 188 and the low resolution of density had resulted in the incorrect positioning of the latter region relative the remaining N_NTD_ (see below).

As in other *Mononegavirales* N^0^−P complexes, the P chaperone module prevented the polymerization of N by interfering with the binding of the N_NT-ARM_ and N_CT-ARM_ subdomains from adjacent N subunits. The N-terminal part of RABV P_68_ competed with the N_CT-ARM_ of the N*i*+1 protomer, whereas the C-terminal part competed with the N_NT-ARM_ of the N*i*–1 protomer (Figure 2H). Additionally, in the RABV N_Δ23_^0^−P_68_ complex, the RNA binding groove in the N protein adopted a close conformation as in the N−RNA complex [39] and in VSV N_Δ21_^0^−P_60_ structure [40] (Figure 2F). The structural alignment with VSV N_Δ21_^0^−P_60_ structure showed that the overall orientation of P_68_ relative to N was similar to that of P_60_ in VSV complex, although its exact position on the surface of N, notably its C-terminal part, was slightly different (Figure 2J). Finally, as in the VSV N_Δ21_^0^−P_60_, the RABV P_68_ protruded into the RNA binding groove of N and interfered directly with RNA binding (Figure 2I and see below).

### 3.3. The Chaperone Module Forms a Compact but Fuzzy Complex with N_Δ 23_^0^

To obtain structural information about the disordered parts of the RABV N_Δ23_^0^−P_68_ complex, we turned to SAXS and SEC-SAXS. We injected a sample of N_Δ23_^0^−P_68_ on a Superdex 200 Increase column and collected SAXS data at regular intervals along the elution peak (Figure 3A). We also collected SAXS curves in batch mode over a concentration range from 3.0 to 10.0 mg.mL^−1^ (scattering vector (q) values ranging from 0.06 to 5.3 nm^−1^) (Figure 3B). The shapes of the scattering curves were independent of protein concentration (Figure 3B). The Guinier plots at low q values (q.R_g_ < 1.4) were linear (Figure 3C) and the R_g_ value calculated by using Guinier approximation was constant throughout the chromatographic peak in the SEC-SAXS experiment (Figure 3A) and showed no dependence on the protein concentration (Figure 3D), thus indicating the absence of aggregation or intermolecular interaction. The dimensionless Kratky plots reached a plateau near 1.2 for q.R_g_ values near 1.7, which is indicative of a globular structure (Figure 3E). Again, in an effort to optimize the signal-to-noise ratio and minimize the possible structure factor contribution, we merged the curves at the different concentrations [77] (Figure 3F) and used the software GAJOE to further explore the conformational diversity of the complex with the Ensemble Optimization Method (EOM). [70]. We generated an initial ensemble of ~6600 atomic models of the N_Δ23_^0^−P_68_ complex by molecular dynamics simulations (see below) and used the software GAJOE to select sub-ensembles of three conformers that reproduced the experimental SAXS curve ($\chi_{\exp}^{2}=0.95$, Figure 3F). The R_g_ and D_max_ distributions of these selected sub-ensembles indicated the presence of two main populations of the N_Δ23_^0^−P_68_ complex (Figure 3G,H). One population (50%) corresponded to conformers where the C-terminal part of P_68_ was located in the RNA binding groove and bound to the surface of N (conformers 1 and 2 in Figure 3I). This is in agreement with the presence in the flexible part of P_68_ (aa 40–68) of eight acidic residues (Asp + Glu) for only three basic (Lys + Arg) and the presence of basic residues in the RNA binding groove of N. This insertion in the RNA binding groove could also contribute to the chaperon activity of P by competing with RNA molecules (Appendix A). The other population (50%) corresponded to conformers where the C-terminal part of P_68_ extended in the solvent (conformers three in Figure 3I) and was rather extended in comparison with the statistical distribution of the initial ensemble (Figure 3G,H).

### 3.4. A Slow Off-Rate Sets the Affinity of RABV P Chaperone Module for N^0^ in the Low Nanomolar Range

The interaction between RABV N_Δ23_^0^ and P_68_ resisted separation by size exclusion chromatography (Figure 2B), suggesting by a rule of thumb that the dissociation constant was less than 1 μM. However, our inability to prepare N_Δ23_^0^ protein in an isolated, monomeric form prevented the easy measurement of the affinity for P_68_. In order to quantify this interaction in solution, we developed a competition fluorescence anisotropy assay to monitor the reaction (Figure 4A). We generated, expressed, and purified a variant of P_42_ where we replaced the glycine residue at position 41 by a cysteine (P_42-G41C_) and we chemically labeled this cysteine with fluorescein (FAM). We showed by mass spectrometry that the labeling was complete at more than 95% (Appendix A). We reconstituted the complex with the fluorescently labeled peptide (N_Δ23_^0^−P_42−G41C_*), purified it by SEC, and confirmed by SEC-MALLS analysis that it formed a heterodimer. At a concentration of 100 nM of N_Δ23_^0^−P_42-G41C_* complex, fluorescence anisotropy was near 0.24, whereas that of the isolated peptide P_42-G41C_* was near 0.07 (Figure 4B). We measured the dissociation kinetics at 20 °C in the absence and presence of an unlabeled competing peptide. In the absence of competing peptide, the anisotropy value remained stable for more than 24 h (Figure 4B), supporting a low nanomolar range affinity constant. In the presence of a large excess of unlabeled competing P_42_ or P_42-G41C_ (final concentrations of 15 μM), we observed a monotonous decrease in fluorescence anisotropy revealing that the fluorescent peptide dissociated from N_Δ23_^0^ and that the system evolved toward a new equilibrium (Figure 4B). In a first approach, assuming an irreversible first-order dissociation process, we fitted the dissociation curve in the presence of unlabeled peptide with a single exponential equation and obtained a k_off_ value of 0.0027 ± 0.0001 min^−1^ (half time = 256 min). In a second approach, we assumed similar k_on_ and k_off_ rate constants for P_42-G41C_* and P_42-G41C_ and fitted the dissociation curve in the presence of the competitor peptide to a reversible exchange mechanism (Figure 4A) by using a numerical integration method implemented in the software Dynafit [55] and obtained k_off_ = 0.0027 ± 0.0001 min^−1^ (45.10^−6^ s^−1^) and k_on_ = (5.7 ± 0.1) 10^−3^ nM^−1^ min^−1^ (95,000 M^−1^ s^−1^). The dissociation constant K_d_ of 0.50 ± 0.03 nM was readily calculated from these rate constants (k_off_/k_on_). However, in the conditions used in our experiment, the kinetics were dominated by the off-rate as shown by the identical k_off_ value obtained in both fitting procedures. Thus, the value of k_on_ should be considered a lower bound value, as no significant difference in the goodness of fit was found when larger values were imposed and the K_d_ value of 0.50 nM represented an upper bound value. Considering an upper limit for a bimolecular protein–protein interaction on-rate (k_on_) near 5 × 10^5^ M^−1^ s^−1^ [78], a lowest bound estimate for K_d_ value of 0.09 nM was obtained.

We then measured equilibrium competition binding curves by incubating the FAM-labeled N_Δ23_^0^−P_42-G41C_* complex with serially diluted concentrations of unlabeled P_42-G41C_ and P_68_. The optimal incubation time required for the system to reach equilibrium, mainly depending on the off-rate (k_off_), was set by a rule of thumb at 24 h (5 × 0.693/k_off_ = 1284 min = 21 h). The curve provided a measure of the relative affinity of unlabeled peptide to labeled peptide. All things being equal, we expected that the half-saturation concentration should be that of the initial concentration of the labeled complex (C_1/2_ = 100 nM indicating that in a solution containing 100 nM labeled peptide and 100 nM unlabeled peptide, the protein was 50% saturated by the labeled peptide). The curve obtained with unlabeled P_42-G41C_ had a midpoint (C_1/2_) around 300 nM, whereas that with P_68_ had a midpoint near 1000 nM (Figure 4C), revealing that the FAM moiety slightly contributed to the stability of the complex (previous attempts with P_42_ labeled with FAM at its N-terminus revealed an even stronger stabilization) and that the C-terminal part of P_68_ slightly reduced the stability of the complex. If we assumed K_d_* = 0.5 nM for the labeled peptides, fitting the equilibrium competition binding curves by numerical integration with the software Dynafit [55] yielded K_d_ values of 4.5 ± 0.5 nM (ΔΔG = +1.3 kcal/mol) for P_42-G41C_ and of 8.2 ± 0.6 nM (ΔΔG = +1.7 kcal/mol) for P_68_ (Figure 4C and Table 2).

To further characterize how individual residues in P_68_ contributed to the binding affinity for N_Δ23_^0^, we analyzed the structure with FoldX, a knowledge-based algorithm for predicting hot spots in protein–protein interfaces [52]. Four residues were predicted to play a key role in the stability of the complex with ΔΔG > 2 kcal/mol for Ala replacement: Phe5, Pro8, Val25, and Ile28, whereas seven other residues could be considered as milder contributors with 2 kcal/mol > ΔΔG > 1 kcal/mol: Ile4, Arg12, Glu22, Thr24, Leu27, Ile32, and Asn35. To test these predictions and validate our structural model, we generated, produced, and purified alanine variants for four of these residues (Phe5Ala, Pro8Ala, Arg12Ala, and Ile28Ala), including as a control the variant Arg12Ala, which was not predicted to be a hot spot and measured equilibrium competition binding curves (Figure 4D and Table 2). The curve obtained with the mutant P_42,G41C,R12A_ was close to that obtained with P_42,G41C_, with a K_d_ value of 2.9 ± 0.6 nM. By contrast, the three other mutant peptides were less efficient than the wt peptides for displacing the FAM-labeled peptide and, consequently, the equilibrium binding curves were shifted to higher peptide concentrations, confirming the role of these residue side chains in stabilizing the complex. The results confirmed the importance of these key residues in the stabilization of the complex and the order of experimental ΔΔG values reflects the predictions of FoldX (Figure 4D and Table 2).

### 3.5. Binding of the Chaperone Module of RABV P and RNA-Mediated Polymerization of N Restrict Motions in N and Stabilize the Protein in Distinct Conformational States

In known structures of the N^0^−P_CM_ complex, N was locked either in an open (*Paramyxoviridae*, *Pneumoviridae*, and *Filoviridae*) [26,28,43,44,45,46,47,79] or a closed conformation (*Rhabdoviridae*) ([40] and this paper). In *Rhabdoviridae*, the closed conformation seen in the N^0^−P_CM_ complex is almost identical to the conformation observed in the polymeric, RNA bound state [39,41]. We hypothesized that P_CM_ interfered with the dynamics of the N protein as part of the chaperoning mechanism and, in particular, perturbed the opening and closing of the RNA binding groove [43]. To test this model, we used N_Δ23_^0^ alone (extracted from the crystal structure of N_Δ23_^0^−P_68_), N_Δ23_^0^−P_68_, and N_3_−RNA, a set of three adjacent subunits extracted from the circular N_11_−RNA complex (2GTT.pdb), in which we built the flexible parts to generate full-length N molecules. We performed molecular dynamics (MD) simulations to evaluate the effects of P_68_, RNA binding, and multimerization on the dynamics of N protein (Appendix A). The N_Δ23_^0^ and N_Δ23_^0^−P_68_ systems were stable over the course of the simulations, although high flexibility was observed in loop regions, particularly in the C-terminal loop (N_CT-ARM_) (Appendix A). For N_Δ23_^0^−P_68_, the C-terminal part of P_68_ (res from 42 to 68 + tag) remained flexible and sampled a wide range of conformations (Appendix A), consistent with the SAXS-based ensemble analysis (Figure 3). In the MD simulations of the N_3_−RNA system, the C-terminal domains remained stable as previously found [80], but the N-terminal domains started to unfold in the first 100 ns of the simulation (Appendix A), reflecting possible packing defects due to the incorrect construction of the protein segment from residue from 135 to 200 (see above). We thus re-refined the N_11_−RNA model against the original dataset using more recent software [61,63,64] and the model of N extracted from the N_Δ23_^0^−P_68_ structure (built from higher resolution data), which yielded an improved model of the circular undecameric N−RNA complex (Figure 5 and Table 1). Using three adjacent N subunits (N_3_−RNA) from the re-refined N−RNA complex structure, the MD simulations showed that the N protein structure was stable over several hundred ns in multiple independent trajectories (Appendix A). The root mean square deviation (RMSD) values stabilized around 0.6–0.8 nm after an initial rise during the first 100 ns.

In order to understand how P_CM_ binding, RNA binding, and multimerization modulated the conformational landscape of the N protein, we performed a principal component analysis (PCA) on the MD simulation datasets, using the N residues common to all three systems (Figure 6A). The principal components analysis (PCA) revealed that about 45% of the variance is accounted by the first two principal components, while about 70% of the variance was accounted by the first eight components (Figure 6B). The first component (PC1) represented a twist or shear motion of N_NTD_ relative to N_CTD_ around the hinge region, whereas the second component (PC2) corresponded mainly to the opening−closing movement of the RNA binding groove also around the hinge region (Figure 6C). The 2D projections of the two first principal components indicated that N^0^ alone was more dynamics than N^0^ in complex with P_68_ or than in the N−RNA complex, as evidenced by the larger basin sampled during MD (Figure 6A). Strikingly, we found that N^0^ mainly populated a region in PC2 space characterized by −4 ≤ PC2 ≤ 0 (Figure 6A, left panel), which corresponded to a much more open state relative to N^0^−P_68_ or N−RNA (−1 ≤ PC2 ≤ 3) (Figure 6A, middle and right panels). N^0^ also sampled a large range of values for PC1 (−4 ≤ PC1 ≤ 2) with a main basin centered around PC1 ≈ from −2 to −1, while PC1 values for N−RNA were comprised in a shorter range between −3 and 0. This suggested that RNA binding and multimerization moderately impacted interdomain twisting of N as both systems showed PC1 values roughly centered around the same value (Figure 6C). By contrast, the PC1 values for N^0^−P_68_ were mainly comprised between 0 and 3, implying that P_68_ binding twisted the relative orientation of N_NTD_ and N_CTD_ through its intermolecular contacts in the hinge region (Figure 6C).

In conclusion, the presence of P_68_, as well as RNA and multimerization, restricted the dynamics of N and induced the closure of the RNA binding groove. The P_68_ binding further stabilized the N protein in a twisted conformation relative to N^0^ or N−RNA. These data were consistent with an encapsidation mechanism in which the release of P_CM_ unlocked the opening−closing movements in the N protein, allowing the protein to grab the RNA.

### 3.6. Dimeric Full-Length RABV P Can Chaperone Two N^0^ Molecules

The RABV phosphoprotein forms dimers [17,19,20,22]. To determine whether each chain in a dimeric P molecule was able to chaperone one N^0^ molecule as previously shown for VSV P [42], we reconstituted the complex with full-length P (P_FL_) and purified the N_Δ23_^0^−P_FL_ complex by SEC on a Superdex 200 column. We performed a SEC-SAXS experiment by injecting 50 μL of a sample of N_Δ23_^0^−P_FL_ into a Superdex 200 Increase column and monitored the elution by SAXS. We collected scattering profiles for q values ranging from 0.085 nm^−1^ to 3.0 nm^−1^ at regular time intervals during the elution. The complex eluted as a single peak, but the radius of gyration (R_g_) determined by using the Guinier approximation at small scattering vector (q) values (q_max_.R_g_ < 1.3) varied across the elution peak (Figure 7A) and the scattering intensity profiles in the front part (F1) and the tail part (F2) of the peak were slightly different (Figure 7B, upper panel), suggesting heterogeneity of the eluting species. The differential scattering intensity profile (Figure 7B, lower panel) indicated that the main differences between F1 and F2 were in the low and medium q ranges. The difference at low q was likely due to both differences in protein concentration (the protein concentration was not measured during elution and the profile could not be normalized) and in the molecular mass of the eluting particle (molecular heterogeneity). The negative band between 0.2 and 0.8 nm^−1^ in the differential scattering intensity profile corresponding to distances between 1 and 5 nm in the real space could result from the presence of an additional N molecule.

The Guinier plots for F1 and F2 were linear and we determined R_g_ values of 5.07 ± 0.02 nm for F1 and 4.69 ± 0.02 nm for F2 (Appendix A) from the slope and molecular mass (MM) values of 150 ± 10 kDa for F1 and 120 ± 10 kDa for F2 from the invariant V_c_. The MM determined for F1 was closed to the theoretical mass of 165 kDa calculated for a 2N−2P complex and that for F2 was closed to the theoretical mass of 116 kDa calculated for a 1N−2P complex, also supporting the presence of complexes were each dimer of P bound two (F1) or one (F2) N molecules.

To further investigate this hypothesis, we generated two ensembles of physically accessible conformers of the N_∆23_^0^−P_FL_ complex_,_ one ensemble with one N attached to each chain of P (2N−2P) and one with a single N attached to one of the P chain (1N−2P). We then used the GAJOE program for selecting sub-ensembles that reproduced the experimental SAXS curves at positions F1 and F2 in the chromatogram. The curve in the front part of the elution peak (F1) was better reproduced with 2N−2P models (reduced χ^2^ = 1.9) than with 1N−2P models (reduced χ^2^ = 2.7) (Figure 7C and Appendix A). A representative sub-ensemble of four different conformers in different proportions is shown in Figure 7D. The curve in the tail part of the elution peak (F2) was better reproduced with 1N−2P models (reduced χ^2^ = 1.9) than with 2N−2P models (reduced χ^2^ = 19.9) (Figure 7F and Appendix A). A representative sub-ensemble of three different conformers in different proportions is shown in Figure 7G. The selected ensembles also revealed that both forms of the complex (2N−2P and 1N−2P) were rather compact, with most conformers in the selected ensembles with R_g_ value smaller than the mean R_g_ of the initial distribution (Figure 7E,H). Surprisingly, the R_g_ values of the complexes at positions F1 and F2 were very similar to the R_g_ value of 4.9 ± 0.1 nm determined previously for RABV P alone [17,19]. Furthermore, we calculated that a 2N−2P complex, where the two N molecules were assembled side-by-side as in the N−RNA complex, would introduce features in the scattering intensity curve that were not observed in our experiments, providing evidence that both N^0^ molecules remained independent within the complex.

## 4. Discussion

### 4.1. The N-Terminal Region of P Contains an Autonomous Chaperone Module (P_CM_)

As previously demonstrated for RABV and other viruses, the N-terminal intrinsically disordered region of P contains the chaperon module (P_CM_) that autonomously maintains the N protein (N^0^) in an RNA-free, monomeric, and soluble form [23,25,26,27,28,40,43,44,81]. As found for VSV and NiV, an isolated peptide comprising the RABV chaperone module is sufficient for reconstituting a heterodimeric complex with an armless N^0^ (RABV N_∆23_ lacking the N_NT-ARM_) [40,43]. However, if P_CM_ is sufficient to keep N^0^ in a soluble, RNA-free monomeric form, it is possible that NC assembly in cells requires additional functional modules of P. Indeed, exogenous peptides encompassing P_CM_ from different viruses, including RABV, have been found to inhibit viral replication in cultured cells, whereas they were able to solubilize N in the cytoplasm [24,28,43,79,81], suggesting that their antiviral activity is based on the sequestration of the N protein in unproductive complexes or in a wrong cellular compartment. The C-terminal part of P attaches to the NC [21,80] and the N-terminal region binds to the L-RdRP [82], and either one of these interactions could be necessary to recruit the unassembled N^0^ molecules near the site of RNA synthesis and/or to trigger NC assembly. The dimerization of P through its multimerization domain (P_MD_) [22] was found to be unessential for the activity of the polymerase [83], although it was essential for formation of the membraneless compartments where viral replication occurs [9]. Full-length RABV P might thus be required to recruit N^0^ within the biological condensates, in which viral RNA replication occurs. Finally, full-length RABV P protein forms dimers [19,22], where each chaperone module behaves independently of the other allowing the binding of one or two N^0^ molecules per P dimer depending on the concentration of N present at the time of the reconstitution of the complex [42].

Surprisingly, the overall dimensions of the N_∆23_^0^−P_FL_ complex are not different from those of isolated P_FL_ in similar experimental conditions [17,19]. The scattering intensity at the zero angle at different positions in the chromatographic peak clearly indicates that one or two N^0^ molecules were bound to the P dimer, but the radii of gyration are similar. The SAXS data also clearly show that the two bound N proteins are not preassembled within the N^0^−P complex, but rather remain independent of each other. It is worthwhile noting that if we assume a generic partial specific volume of 0.73 mL.gr^−1^ for both RABV P and N proteins [84] and consider the volume defined by the radius of gyration, the isolated P protein occupies 17% of the volume (R_g_ = 4.9 nm, MM = 68.4 kDa), whereas the 2N−2P complex occupies 36% of the volume (R_g_ = 5.1 nm, MM = 165 kDa) (Figure 8). Similar calculations indicate that VSV P and NiV P occupy 13% and 7% of the volume defined by the radius of gyration, respectively [85,86]. The attachment of N molecules on P thus lead to an increase in density with almost no change in the dimensions of the protein complex as compared to P protein alone.

Additionally, as found for other viruses [26,76], the truncated N protein (N_∆23_) was unable to assemble into NC even in the presence of RNA. The assembly of N into polymeric N−RNA complexes could only be reproduced in vitro with full-length N, demonstrating that the N_NT-ARM_ is required for NC assembly (Bourhis and Jamin, unpublished data), although the mechanism of NC assembly and the detailed role played by the N_NT-ARM_ and N_CT-ARM_ remain poorly understood. We hypothesize that for an incoming N protein to incorporate into a nascent NC, the binding of its N_NT-ARM_ onto the last already incorporated N subunit could be required either for anchoring the new N protein and allowing the assembly or for stabilizing the complex once it is formed or both.

### 4.2. Thermodynamic Control of NC Assembly

The interaction between N_∆23_^0^ and P_68_ is strong with a dissociation constant in the low nanomolar range. The dissociation constant of the FAM-labeled peptide was below 1 nM and that of the unlabeled peptide measured by competition assay was 4 nM. This strong affinity is consistent with the large amount of surface area that becomes buried upon association with P_68_ (~2800 Å^2^). The binding of the N-terminal part of P_CM_ (aa 4–15) onto N_∆23_^0^ buries about 1200 Å^2^, whereas the binding of the C-terminal part (aa 20–38) buries about 1600 Å^2^. For protein–protein and protein−peptide complexes that bury more than 2000 Å^2^ upon assembly, an average value of 4.0 cal/mol/Å^2^ was estimated for the ∆G_binding_ [87], which for the burial of 2800 Å^2^ would predict a ∆G_binding_ of 11.2 kcal.mol^−1^ and a dissociation constant of 6 nM (at 20 °C), which is in good agreement with our measurements. By comparison, a dissociation constant of 1.1 nM has been determined for the interaction between Ebola virus N^0^ and a peptide of VP35 that contains the chaperone module, whose association buried a surface area of ~2400 Å^2^. These figures place RABV N^0^−P complex among the protein–protein complexes that bury the largest surface area upon assembly and have a high binding affinity [87]. This strong interaction might be required for assembling the N^0^−P complex in the infected cells at the earliest stages of the virus replication cycle, when only small amounts (low concentrations) of viral proteins are present. The nucleoprotein has a strong tendency to nonspecifically assemble onto cellular RNA and thus chaperoning by the P protein must occur as soon as N synthesis begins. However, the N^0^−P complex is not the final step in the production line of the viral N protein. N^0^ must be subsequently transferred to nascent viral RNA to form new viral NCs. To drive the assembly of NC, the N−RNA must probably be as stable or even more stable than the N^0^−P complex and it is noteworthy that binding of the N_NT-ARM_ of the N_i-1_ subunit buries ~1500 Å^2^, while the binding of the N_CT-ARM_ of the N_i+1_ subunit buries ~1700 Å^2^, totaling ~3200 Å^2^ per N subunit upon NC assembly, which translates into predicted ∆G_binding_ value of 12.8 kcal.mol^−1^ and a dissociation constant of 0.3 nM (at 20 °C). The assembly of NC from the N^0^−P complex could thus simply be driven by thermodynamics, the reaction evolving toward the most stable state of the system.

The molecular dynamics simulations revealed that the C-terminal part of P_68_ can transiently insert into the RNA binding groove of N perturbing/preventing the insertion of an RNA molecule and thus provide a putative additional contribution to the N^0^ chaperon activity of P. This part of RABV P_NTD_ is rich in negatively charged amino acid and is thus suitable for inserting in the positively charged RNA binding groove. However, this part of the peptide is not visible in the crystal structure, indicating either that the interaction is weak and transient or that the peptide binds in different conformations in the RNA binding groove. In the meantime, the presence of this region destabilized the complex, possibly because the decrease in conformational entropy is not compensated by the formation of additional stabilizing interactions.

### 4.3. The Chaperone Module of P and the N_NT-ARM_ of the Adjacent N Protein Controls the Opening and Closing Movement in N

A comparison with the other known structures of the N^0^−P_CM_ core complex from different viruses belonging to different families of the *Mononegavirales* revealed many similarities in the complex architecture and in the mechanism of action of P_CM_, but also some differences [29,40,43]. Globally, all these complexes have a contiguous segment of about 30–35 residues located at or near the N-terminal end of the P protein that binds mainly or solely to N_CTD_ and directly competes with the N_NT-ARM_ and N_CT-ARM_ of adjacent subunits. In the *Paramyxoviridae* (NiV, MeV, hPIV3, PIV5), *Pneumoviridae* (hMPV), and *Filoviridae* (EBOV and MARV), P_CM_ binds exclusively to N_CTD_, in the same groove as the N_NT-ARM_ subdomain of the N_i-1_ subunit in the polymeric N−RNA complex. In these complexes, the two protein segments (P_CM_ and N_NT-ARM_) adopt opposite orientations although they bind in the same groove of the N protein [26,43,44,45,46]. In some cases, P_CM_ extends to the binding of the N_CT-ARM_ of the N_i+1_ subunit and thus also directly interferes with its binding to the N_i_ subunit. In the *Rhabdoviridae* (VSV and RABV), one part of P_CM_ also binds to N_CTD_, in the same groove as the N_NT-ARM_ of the N_i-1_ subunit and the N_CT-ARM_ of the N_i+1_ subunit, while the major part of the chaperone module forms an α-helix that binds at the interface between N_NTD_ and N_CTD_ (Figure 2). In the *Rhabdoviridae*, we confirm here that in RABV N^0^−P complex, as previously shown in the VSV complex, the polypeptide chain of P_CM_ follows the same orientation than that of the N_NT-ARM_ [40].

The first crystal structures of polymeric, circular N−RNA complexes revealed that the RNA molecule was embedded in its protein shell, with some bases facing the protein away from the solvent [39,41,88]. This directly suggested that the N protein must open and close, with the N_NTD_ and N_CTD_ moving relative to each other by rotation around a hinge located in their connecting region. This motion not only allows the insertion of the RNA molecule during the assembly of the NC but also provides access to all the nucleotide bases during the passage of the polymerase. Another striking difference observed by comparing the N^0^−P_CM_ complexes of different viruses was the conformation of the N protein. In the N^0^−P_CM_ complexes of *Rhabdoviridae* (VSV and RABV), the N protein is locked in a closed conformation, causing the RNA binding site to be inaccessible to an incoming RNA molecule and suggesting that the protein must open to accommodate an RNA molecule. When initially observed in the VSV N^0^−P_CM_ structure, the polymeric assembly of N raised the possibility that the closure of the RNA binding groove was a consequence of the polymerization [40]. In the new crystal structure of the RABV N^0^−P_CM_ complex, the N protein is unassembled but is also trapped in a similar closed conformation, demonstrating that this state of the protein is independent of the polymerization and is an intrinsic property of the protein–protein complex as confirmed by our principal component analysis of MD simulations.

By contrast, in the N^0^−P_CM_ complexes of *Paramyxoviridae*, *Pneumoviridae*, and *Filoviridae*, the N protein is blocked in an open conformation, in which the RNA-binding groove is too widely open to allow the formation of all the bonds evidenced between the protein and the RNA molecule within the N−RNA complexes [43,44,45,46,88,89,90]. In these cases, the protein is rather in a conformation ready to receive an incoming RNA molecule. This finding with a visual analysis of the NiV N^0^−P_CM_ complex led us to propose that the binding of P_CM_ perturbs the dynamics of the N protein by linking together different subdomains that must move relative to one another when the protein switches between its open and close conformations [29,43]. The MD simulations reported in this paper provide evidence that the N protein has a natural tendency to undergo large conformational changes, where the N_NTD_ and N_CTD_ move relative to each other in different directions. The binding of P_CM_ but also of the N_NT-ARM_ and RNA onto the surface of N restricts these movements in the protein. The binding of P_CM_ blocks N in open or close conformation depending on the viral family, preventing the positioning of an RNA molecule in the groove between N_NTD_ and N_CTD_. The binding of RNA, N_NT-ARM_, and also N_CT-ARM_ blocks N in a close conformation that shields the RNA from the solvent, cellular proteins, and the viral polymerase. Different hypotheses have been put forward to explain the opening of the N protein, involving the intervention of P or L to trigger this process. The simulations suggest that the release of P_CM_ and of the N_NT-ARM_ from the surface of N would allow the protein to undergo its intrinsic opening−closing movements and allow the insertion or release of the RNA molecule.

## 5. Conclusions

On the basis of the structural architecture and tentative mechanism of action of the N^0^−P_CM_ complex, viruses can currently be separated into two different groups, with one group including the *Rhabdoriviridae* and another group including *Paramyxoviridae*, *Pneumoviridae*, and *Filoviridae*, which likely reflect evolutionary relationships. The formation of the N^0^−P complex is likely an ancestral feature of the *Mononegavirales*, which has evolved by divergence in the different families, conserving the essential properties and showing that the great sequence−structure space diversity of the polypeptide chain allied with the high capacity to mutate of these viruses can create important variations around the same “theme”. A striking example is the inversion of the direction of binding of the P_CM_ on N^0^.

## Figures and Tables

**Figure 1 viruses-14-02813-f001:**
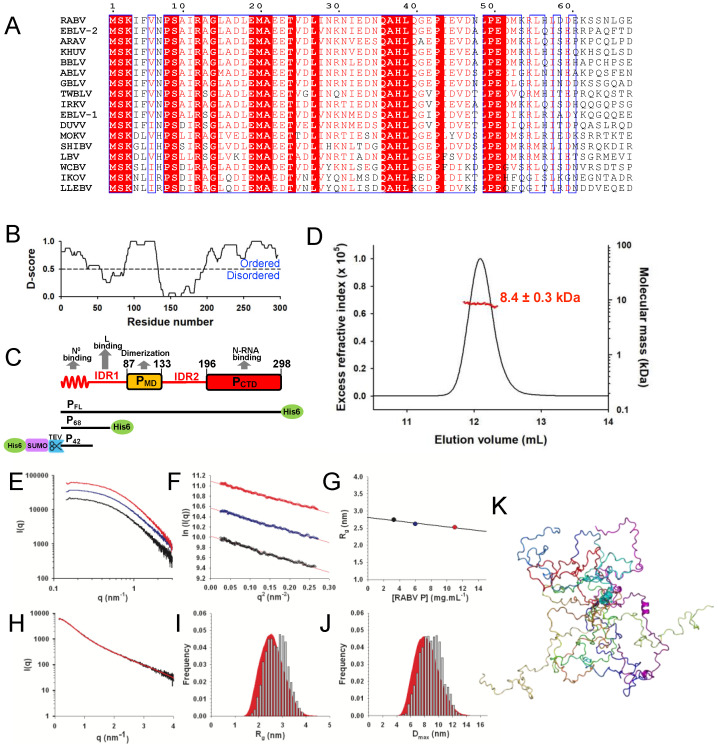
The N^0^ chaperone module of RABV phosphoprotein. (**A**) Multiple sequence alignment of Lyssavirus P_NTR_ region. Members of the Lyssavirus genus and their UniProt accession number: RABV—rabies virus CVS−11 strain P22363, EBLV−1—European bat lyssavirus 1 A4UHP9, EBLV−2—European bat lyssavirus A4UHQ4, ARAV—Aravan virus Q6X1D7, KHUV—Khujand virus Q6X1D3, BBLV—Bokeloh bat lyssavirus U3MZL8, ABLV—Australian bat lyssavirus Q9QSP3, GBLV—Gannoruwa virus A0A1J0RI70, TWBLV—Taiwan bat lyssavirus A0A3P8MNG3, IRKV—Irkut virus Q5VKP5, DUKV—Duvenhage virus O56774, MOKV—Mokola virus P0C569, SHIBV—Shimoni bat virus D4NRJ9, LBV—Lagos bat virus D4NRK4, WCBV—West Caucasian bat virus Q5VKP1, IKOV—Ikoma virus J5JWQ7, LLEBV—Lleida virus A0A1I9RG27. (**B**) D-score. A consensus disordered prediction (D-score) was calculated as described in [20]. The threshold to distinguish between the ordered and disordered region was arbitrarily set at 0.5. The shaded areas indicate the positions of the known folded dimerization domain (P_MD_) and NC-binding C-terminal domain (P_CTD_) (see panel (**C**)). (**C**) Schematic representation of RABV phosphoprotein modular organization and constructs. The upper part shows the structural organization of the phosphoprotein. Boxes indicate the localization of folded domains, undulated lines the localization of predicted MoRE and lines the localization of intrinsically disordered regions (IDR 1−2). The grey arrows indicate the location of the regions associated with functions in RNA synthesis. The lower part shows the three constructs of the phosphoprotein used in this study, indicating the position of the tags and cleavage sites. (**D**) Size exclusion chromatography and multiple-angle laser light scattering (SEC-MALLS) of RABV P_68_. The elution was monitored on-line using multi-angle laser light scattering and differential refractometry. The line shows the chromatograms monitored using differential refractive index measurements. The red crosses indicate the molecular mass across the elution peak calculated from static light scattering and refractive index, and the numbers indicate the weight-averaged molecular mass (kDa) with standard deviations. (**E**–**K**) SAXS experiments and modeling. SAXS profiles recorded at 3.3 mg.mL^−1^ (in black), 6.0 mg.mL^−1^ (in blue), and 11.0 mg.mL^−1^ (in red) are shown in direct plot (**E**) and Guinier plots (**F**). (**G**) R_g_ values obtained from the Guinier approximation. (**H**) Merged SAXS curve (in black). The red line shows the fit obtained by the EOM method with an ensemble of 12 conformers (see panels (**J**,**K**)). (**I**) R_g_ distribution. The shaded area shows the R_g_ distribution of the initial ensemble. The bars show the R_g_ distribution of selected ensembles. (**J**) D_max_ distribution. The shaded area shows the D_max_ distribution of the initial ensemble. The bars show the D_max_ distribution of selected ensembles. (**K**) Representative ensemble of 12 conformers selected by GAJOE. Each conformer is shown in a different color.

**Figure 2 viruses-14-02813-f002:**
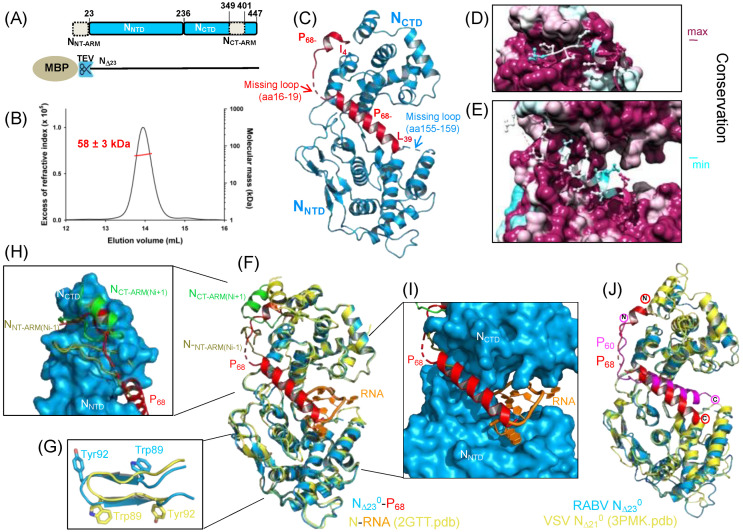
Crystal structure of RABV N_∆23_^0^−P_68_ core complex. (**A**) Schematic representation of RABV nucleoprotein modular organization and construct. The upper part shows the structural organization of the nucleoprotein. Boxes indicate the localization of folded domains, dotted boxes the localization of the exchanging subdomains N_NT-ARM_ and N_CT-ARM_. The lower part shows the construct of the nucleoprotein used in this study. (**B**) SEC MALLS of the reconstituted complex. The elution was monitored on-line using multi-angle laser light scattering and differential refractometry. The line shows the chromatograms monitored by differential refractive index measurements. The red crosses indicate the molecular mass across the elution peak calculated from static light scattering and the numbers indicate the weight-averaged molecular mass (kDa) with standard deviations. Theoretical mass of the heterodimer = 48,208.5 Da (N_Δ23_) + 8488.4 Da (P_68_) = 56,696.9 Da. (**C**) Crystal structure of RABV N_∆23_^0^−P_68_ core complex (8b8v.pdb) in cartoon representation. N_∆23_^0^ is shown in blue and P_68_ in red. The dotted line in P_68_ shows the position of the four residue loop that is less-well defined in the electron density map, while the dotted lines in N show the position of the missing loops in N_NTD_ and the N_CT-ARM_. The N- and C-terminal residues of P_68_ are indicated. (**D**) Conservation of the N^0^−P interface. View of the region of interaction of residues 4 to 15 of P_CM_. The complex is shown with surface and conservation representations for N_∆23_ and with stick-and-ball representations for P_68_. The conservation in N derived from multiple sequence alignment is displayed on the surface of NiV N: blue low-level conservation, <20%; maroon, high-level conservation, >80%. The side chains of conserved residues in the P N-terminal region are shown in stick representation. (**E**) Conservation of the N^0^−P interface. View of the region of interaction of residues from 20 to 39 of P_CM_ with same color scheme and representation as in panel (**D**). (**F**) Superposition of RABV N^0^−P and N−RNA structures. The structures were superposed by aligning the C-terminal domains of both structures. (**G**) Out-of-register construction in N_NTD_. Close-up view of a small region of N_NTD_ illustrating the change of register of residues from 27 to 130 between the original circular N_11_−RNA structure (2GTT.pdb) shown in yellow and the N_∆23_^0^−P_68_ structure (8B8V.pdb) shown in blue. (**H**) Interference of P_CM_ binding with N_NT-ARM_ and N_CT-ARM_ binding. Close-up view of the interference between P_CM_ (in red) and the N_NT-ARM_ from the N_i-1_ subunit (in olive) and N_CT-ARM_ of the Ni+1 subunit (in green). (**I**) Interference of P_CM_ binding with RNA binding. Close-up view of the interference between P_CM_ in (red) and RNA (in orange). (**J**) Superposition of RABV N_∆23_^0^−P_68_ and VSV N_Δ21_^0^−P_60_ complex.

**Figure 3 viruses-14-02813-f003:**
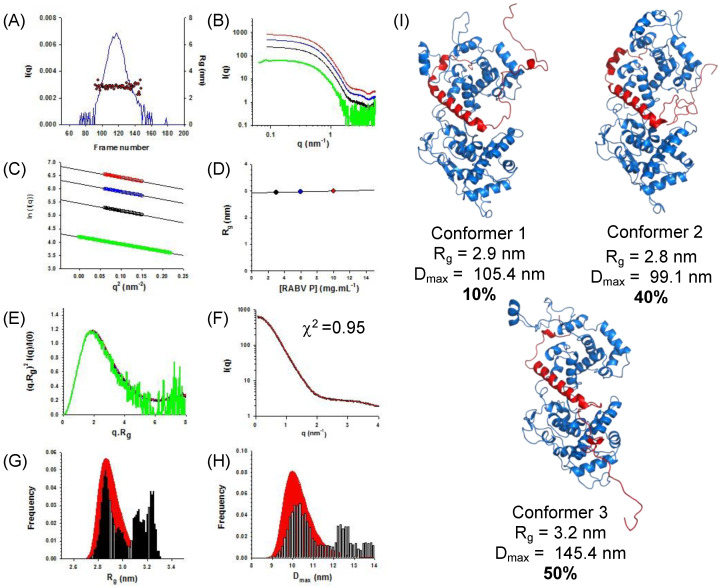
SAXS and SEC-SAXS of RABV N_∆23_^0^−P_68_ core complex. (**A**) SEC-SAXS elution profile and R_g_ across the elution peak; 50 μL of the N_∆23_^0^−P_68_ sample were injected onto a Superdex 200 column and monitored on-line by SAXS. The black line shows the intensity at zero angle (I_0_), which is proportional to both the MM and concentration. The red dots indicate the values of the radius of gyration calculated from the Guinier approximation at the different time intervals. (**B**) SAXS profiles at different protein concentrations. SAXS profiles were recorded in batch mode at 3 mg.mL^−1^ (in black), 6 mg.mL^−1^ (in blue), and 11 mg.mL^−1^ (in red). The curve in green was obtained by averaging the individual profiles recorded throughout the SEC elution peak shown in Panel (**A**). (**C**) Guinier plots at different protein concentrations. Same color scheme as in panel (**B**). (**D**) Rg at different protein concentrations. The R_g_ value were calculated from the Guinier plot (panel (**C**)) for the profiles recorded in batch mode. (**E**) Dimensionless Kratky plots at different protein concentrations. Same color scheme as in panel (**B**). (**F**) Merged curve and conformational ensemble modeling by the ensemble optimization method (EOM). The black line shows the scattering profile obtained by merging segments of the profiles obtained at different protein concentrations (panel (**B**)). The red line shows a back-calculated scattering curve for a selected ensemble of three conformers measured in different proportions and shown in panel (**I**) (χ^2^ = 0.95). (**G**) R_g_ distribution. The red area shows the R_g_ distribution calculated for the initial ensemble of conformers, whereas the black bars show the R_g_ distribution of the selected ensemble that fit the experimental SAXS data (panel (**F**)). (**H**) D_max_ distribution. The red area shows the D_max_ distribution calculated for the initial ensemble of conformers, whereas the black bars show the D_max_ distribution of the selected ensemble that fit the experimental SAXS data. (**I**) Representative ensemble of conformers. N is shown in blue and P_68_ including the eight C-terminal residues of the linker and His-tag are shown in red (cartoon representation).

**Figure 4 viruses-14-02813-f004:**
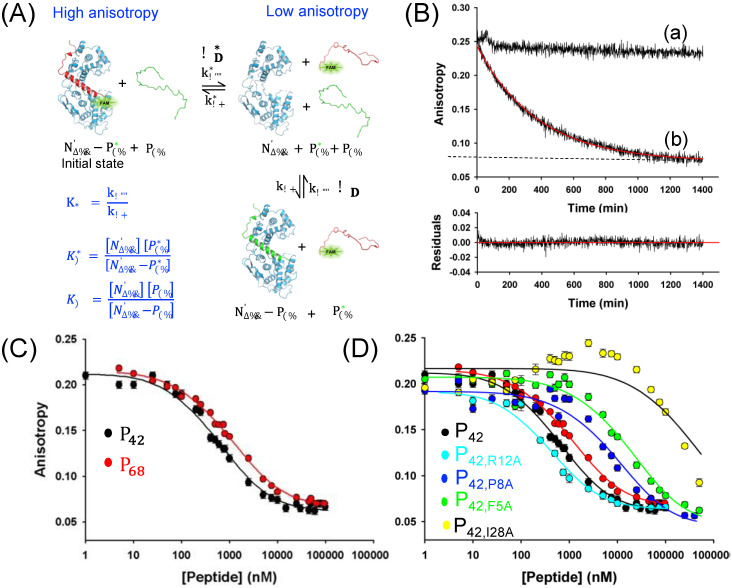
Competitive binding assay using fluorescence anisotropy. (**A**) Schematic representation of the competition assay and the mathematical models. (**B**) Dissociation kinetics; 200 μL of N_∆23_^0^−P_68_ at a concentration of 100 nM in 20 mM Tris/HCl pH 7.5, 150 mM NaCl at 20 °C. The upper panel shows the anisotropy variation in the absence (a) and presence (b) of a competing unlabeled peptide. The average anisotropy of P_68_−FAM alone is indicated by a dotted line. The red line shows the fit obtained with DYNAFIT. The lower panel shows the plot of the residuals for the fit obtained with DYNAFIT and shown in the upper panel. (**C**) Equilibrium binding curves with WT peptides. The black circles are for P_42_ and the red circles for P_68_. The lines show the fits obtained with DYNAFIT and the parameters shown in Table 2. (**D**) Equilibrium binding curves with mutants. The black circles are for P_42_ and are shown as reference. The light blue, dark blue, green, and yellow circles are for the mutant R12A, P8A, F5A, and I28A, respectively, and the lines show the fits obtained with DYNAFIT and the parameters shown in Table 2.

**Figure 5 viruses-14-02813-f005:**
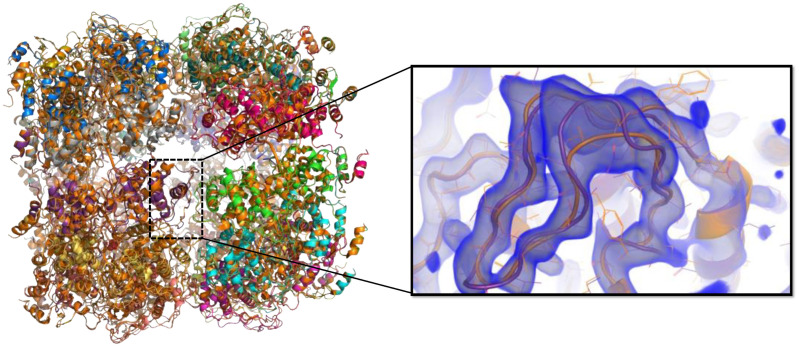
Re-refinement of the circular polymeric N_11_−RNA complex crystal structure. The different N subunits of the re-refined model are shown in cartoon representation with different colors for each chain and superimposed onto the original model (PDB ID 2GTT, shown in orange). The close-up view shows an excerpt of the region that was rebuilt (in orange), with the experimental 2Fo-Fc map shown in volume representation in blue and contoured at 1σ (drawn with PyMOL).

**Figure 6 viruses-14-02813-f006:**
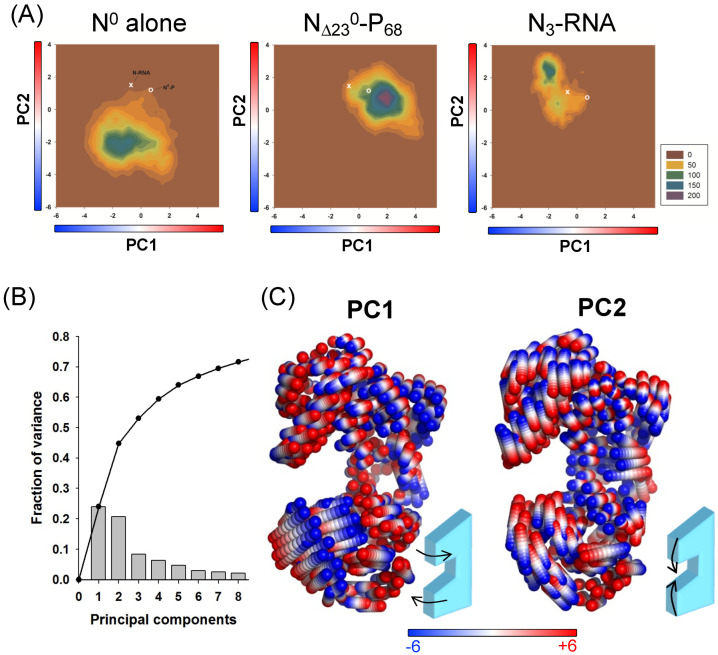
Molecular dynamic simulations. (**A**) Principal component analysis (PCA); 2D projections of the first two principal components (PCs) calculated for the N protein from MD simulations run with N^0^ alone (left panel), N_Δ23_−P_68_ (middle panel) and N_3_−RNA (right panel). The data are shown as a 2D histogram with the density distribution color scheme indicated on the right. The value calculated from the crystal structures of the N_11_−RNA complex and N_Δ23_−P_68_ complex are shown for reference on each graph as a white cross and a white circle, respectively. (**B**) Fraction of the variance captured by each PC (histogram) and cumulative contributions of the first eight PCs. (**C**) Collective motions of N captured by PC1 and PC2. The motions are illustrated as linear interpolations between the extreme projections of the structures onto the PCs. Each cylinder thus describes the path of a Cα atom between its extreme positions (on a red–white–blue color scale).

**Figure 7 viruses-14-02813-f007:**
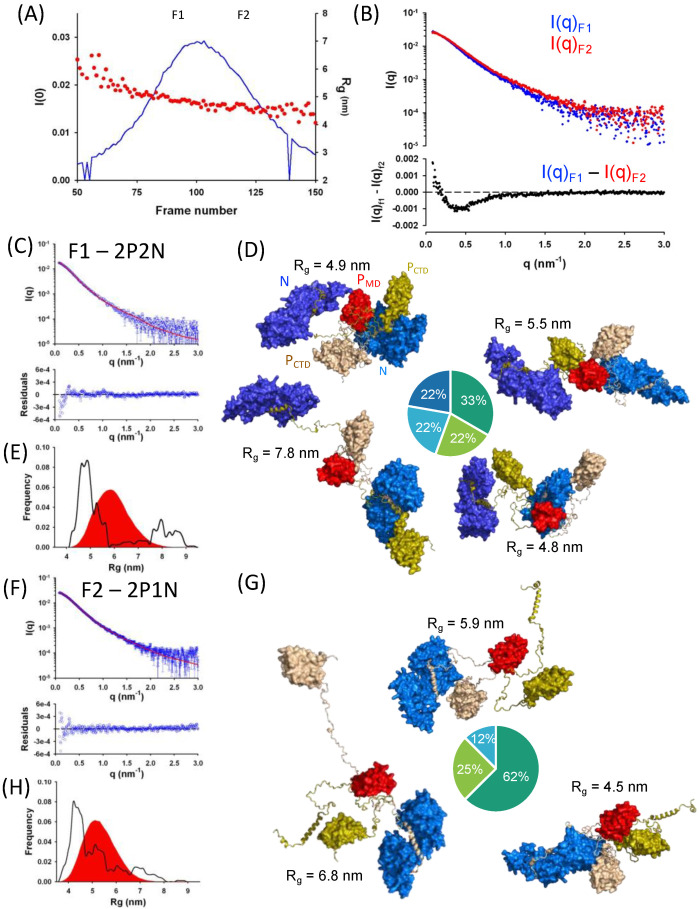
SEC-SAXS of N_∆23_^0^−P_FL_ complex. (**A**) SEC-SAXS elution profile and R_g_ across the elution peak; 50 μL of N_∆23_^0^−P_FL_ sample were injected onto a Superdex 200 column and monitored on-line by SAXS. The blue line shows the intensity at zero angle (I_0_), which is proportional to both MM and concentration. The red dots indicate the values of the radius of gyration calculated from the Guinier approximation at the different time intervals. The shaded areas (labeled F1 and F2) show the frames that were averaged and used for analysis in the next panels. (**B**) Average SAXS profiles at two different positions in the SEC profile. The upper part shows the curves obtained by averaging the individual profiles recorded across the two fractions (F1 and F2) of the SEC elution peak shown in Panel (**A**). The lower part shows the difference scattering profile (I(q)_F1_–I(q)_F2_). (**C**) Conformational ensemble modeling by the ensemble optimization method (EOM). The upper panel show the experimental F1scattering profile (blue circles) The red line shows back-calculated scattering curve for a selected ensemble of 4 2P−2N conformers shown in panel (**E**) (χ^2^ = 1.95). The lower panel shows the plot of the residuals (blue circles). (**D**) Representative ensemble of conformers that reproduce the curve at position F1. The pie chart indicates the fraction of each conformer used in the calculated curve. The dimerization domain of P (P_MD_) is shown in surface representation in red. The rest of the chains of P are shown in wheat and olive and the C-terminal domains (P_CTD_) are shown in surface representation, while the intrinsically disordered regions are shown as cartoons. The N^0^ molecules are shown in two shades of blue in surface representation. (**E**) R_g_ distribution. The red area shows the R_g_ distribution calculated for the initial ensemble of conformers (2P−2N), whereas the black line shows the R_g_ distribution of the selected ensembles that fit the experimental SAXS data (panel (**C**)). (**F**) Conformational ensemble modeling by the ensemble optimization method (EOM). The upper panel show the experimental F1 scattering profile (blue circles) The red line shows back-calculated scattering curve for a selected ensemble of three 2P−1N conformers shown in panel H (χ^2^ = 1.99). The lower panel shows the plot of the residuals (blue circles). (**G**) Representative ensemble of conformers that reproduce the curve at position F2. The pie chart indicates the fraction of each conformer used in the calculated curve. The P protein is shown as in panel E and the N^0^ molecule is shown in blue in surface representation. (**H**) R_g_ distribution. The red area shows the R_g_ distribution calculated for the initial ensemble of conformers (2P−1N), whereas the black line shows the R_g_ distribution of the selected ensembles that fit the experimental SAXS data (panel (**F**)).

**Figure 8 viruses-14-02813-f008:**
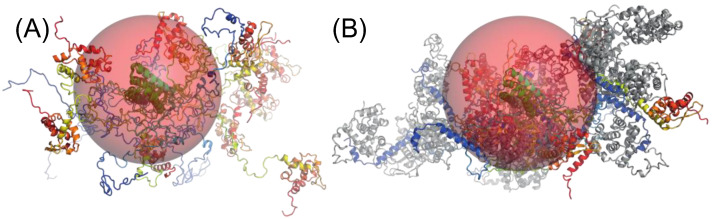
Comparison of ensemble models of P_FL_ and of the 2P−2N N_∆23_^0^−P_FL_ complex. (**A**) Full-length P dimers. Representative ensemble of the five conformers of dimeric P_FL_ selected by GAJOE to reproduce SAXS data [17]. Each P protomer is colored from N-terminus (blue) to C-terminus (red) and the conformers were aligned by superimposing their individual dimerization domain. The semi-transparent red sphere has a radius of 4.9 nm corresponding to the radius of gyration determined from SAXS data. (**B**) 2P−2N N_∆23_^0^−P_FL_ complex. Representative ensemble of the four conformers of the 2P−2N N_∆23_^0^−P_FL_ complex selected by GAJOE to reproduce SAXS data (Figure 7E). Each P protomer is colored from N-terminus (blue) to C-terminus (red), the N_∆23_ molecules are shown in gray and the conformers were aligned by superimposing their individual dimerization domain. The semi-transparent red sphere has a radius of 5.1 nm corresponding to the radius of gyration determined from SAXS data.

**Table 1 viruses-14-02813-t001:** Data collection and refinement statistics.

	N_Δ23_^0^–P_1-68_ (Molecular Replacement)	N_11_–RNA (Re-Refinement)
Data Collection		
Space group	P2_1_2_1_2_1_	P2_1_2_1_2
Cell dimensions		
*a*, *b*, *c* (Å)	39.6, 73.6, 154.3	270.4, 281, 236.9 ^b^
α, β, γ (Å)	90.0, 90.0, 90.0	90.0, 90.0, 90.0 ^b^
Resolution (Å) ^a^	50.0–2.3 (2.44–2.30) ^a^	100–3.49 (3.63–3.49) ^b^
R_merge_ (%) ^a^	12.7% (52.9%)	Not reported ^b^
I/σI ^a^	12.31 (3.76)	12.15 (3.36) ^b^
Completeness (%) ^a^	99.8 (99.5)	99.9 (99.9) ^b^
Redundancy	7.3	Not reported ^b^

Refinement		
Resolution (Å) ^a^	42.17–2.30 (2.36–2.30)	49.75–3.49 (3.62–3.49)
Numbers of reflections ^a^	19,735 (1397)	226,495 (21,091)
R_work_/R_free_ (%) ^a^	16.2/18.7	22.8/25.2 (26.9/31.3 ^c^)
Numbers of atoms		
Protein	3243	73,326
RNA	-	4103
Water	266	0
*B* factors		
Protein	36.2	106.9
RNA	-	105.9
Water	46.3	-
r.m.s deviations		
Bond lengths (Å)	0.010	0.005
Bond angles (°)	1.03	0.87
Ramachandran statistics		
Ramachandran favored (%)	97.3	96.25
Ramachandran allowed (%)	2.7	3.60
Ramachandran outliers (%)	0.0	0.15

^a^ Values in parentheses are for highest-resolution shell. ^b^ Values are reproduced from the original deposition (2GTT.pdb). ^c^ R_work_/R_free_ of the original pdb entry 2GTT are indicated in parentheses.

**Table 2 viruses-14-02813-t002:** Dissociation constants determined by fitting the equilibrium competition binding curves. The K_D_ values were determined by fitting the curves in Figure 4D to the competition model shown in Figure 4A and fixing K_D_* = 0.5 nM. The ΔΔG^0^ values were calculated by reference to the K_D_* value for FAM labeled P_42_ using ΔΔG^0^ = −RT ln (K_D_*/K_D_) and ΔΔG_calc_ are the values predicted by FoldX from the crystal structure.

Variants	K_D_ (nM)	ΔΔG^0^ (kcal.mol^−1^)	ΔΔG_calc_ (kcal.mol^−1^) from FoldX
**P_42,G41C_**	4.5 ± 0.5	1.3 ± 0.1	
**P_42,G41C,F5A_**	150 ± 40	3.3 ± 0.9	4.4
**P_42,G41C,P8A_**	80 ± 20	3.0 ± 0.8	2.4
**P_42,G41C,R12A_**	2.9 ± 0.6	1.0 ± 0.2	1.7
**P_42,G41C,I28A_**	2700 ± 1100	5.0 ± 2.0	3.5
**P_68_**	8.2 ± 0.6	1.7 ± 0.1	

## Data Availability

Coordinates and structure factors have been deposited in the Protein Data Bank under accession codes 8B8V.pdb and the re-refined structure has been deposited under the accession code 8FFR.pdb.

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
