# Peer review of "Structure and Dynamics of the Unassembled Nucleoprotein of Rabies Virus in Complex with Its Phosphoprotein Chaperone Module"

_viruses, 2022, doi:10.3390/v14122813_

Round 1
Reviewer 1 Report
Gérard et, al. studies the structure and dynamics of N and P protein complexes in rabies virus. The crystal structure resembles the previous published VSV NΔ21-P60. Beyond solving the rigid structure from truncated molecules, they did a lot of work analyzing flexibility of the entire complex using both experiment and simulation. From their data and evidence from previous studies, the authors proposed mechanism virus assembly and regulation of RNA binding, which I think is valuable to the field. I recommend the publication of this paper in viruses.
I only have cosmetic suggestions to improve the manuscript.
1. This paper will benefit from taking out unnecessary parts. For example, Lines 436-439 Line 379-381 describe the size exclusion chromatography in detail, which should be move to methods sections instead of appearing in figure legends.
2. Line 89, symbol of alpha is wrong.
3. Line 100, It would be better to define NCT-ARM before using this term.
4. Both figures 2H and 2I are described as “Mechanism of action of PCM”, which could not be true.
5. Line 459: PCM, CM should be subscript.
6. Figure 3I show representative ensemble of conformers, however the percentage 10% 40% and 50% sum up to 100%. Could you explain whether those conformers are representative or exclusive?
7. Figures 7 C-H are not properly cited in the paper.
Author Response
We thank the reviewer for the comments and for pointing out some errors, which have been corrected as described here:
- This paper will benefit from taking out unnecessary parts. For example, Lines 436-439 Line 379-381 describe the size exclusion chromatography in detail, which should be move to methods sections instead of appearing in figure legends.
The sentences were removed from the legends - Line 89, symbol of alpha is wrong.
This was corrected - Line 100, It would be better to define NCT-ARM before using this term.
A sentence has been added to introduce the different parts of the nucleoprotein, including the NCT-ARM - Both figures 2H and 2I are described as “Mechanism of action of PCM”, which could not be true.
We modified the legends to make it clearer - Line 459: PCM, CM should be subscript.
This has been corrected
- Figure 3I show representative ensemble of conformers, however the percentage 10% 40% and 50% sum up to 100%. Could you explain whether those conformers are representative or exclusive?
The presented sub-ensemble sum-up to 100% because the weighted-sum of the contribution from each conformer shown in the figure reproduced the experimental SAXS curve. It is a representative ensemble, because in the Ensemble Optimization Method, when we run the program GAJOE, the program iteratively selects different ensembles of 3 or more conformers in different % and returns the best ensembles. Typically, we obtain multiple sub-ensembles that correctly reproduced the experimental curve and we present here one among them.
- Figures 7 C-H are not properly cited in the paper.
The numbering of the panel in Figure 7 has been changed and the citations of Figure 7C to 7H have been corrected in the MS.
Reviewer 2 Report
During the replication of the viral genome, the RNA synthesis machinery of mononegaviruses encapsidates the nascent RNA with nucleoproteins (N). Before the addition of N onto the genomic RNAs, the viral phosphoprotein (P), the polymerase co-factor, binds RNA-free N monomers (N0), prevents their binding to cellular RNA, and recruits them at the replication site. Several questions remain about the replication and the encapsidation of the viral genomes. For instance, it is still unclear how P avoids the assembly of N on cellular RNA while allowing it on viral RNA during replication.
Gérard and colleagues used different approaches to investigate these questions with rabies virus, an important human pathogen. They solved the atomic structure of rabies N protein (truncated) in complex with an N-terminal segment of P, modeled the disordered regions of P, measured the binding affinity between the two proteins, revealed the N protein can undergo an opening-closing movement, and proposed a model for the encapsidation process. Although some of these observations have already been made on vesicular stomatitis virus, this work provides new information and pushes forward our understanding of these mechanisms.
With this manuscript, the authors present excellent work. The experiments are well-designed and of good quality. The article is well-written and easy to follow.
I only have two suggestions (not mandatory):
1) Since the residues involved in the interaction between P and N have been identified, add a table listing these residues (as supplemental data?).
2) In a paper on parainfluenza virus 5 from Aggarwal and colleagues (Aggarwal et al., 2018, ref. 44 in this paper), the authors used molecular dynamics to look at the opening-closing movement and show that the N protein at the 3’end of the nucleocapsid seems to open more readily than the others. Since Gérard et al. also performed MD simulation on a small nucleocapsid and propose that the absence of the NNT-ARM could favor the opening of N, it would be interesting to compare the movement of the three RABV N protomers of the small nucleocapsid, see the differences in their movements, and compare with PIV5.
Minor comments:
- In the legend of Figure 1D, specify “red crosses” since one cannot distinguish the shape of the crosses.
- Idem for Figure 2B (first, put the crosses in red like for Figure 1D).
- “(F)” is missing in Figure 2.
- Homogenize the names of the protein domains (e.g.: PCM and PCM, NT-ARM and NTARM…)
- Figure 4D, an issue with colors: yellow dots and red line
- Unknown symbol in the legend of Figure 4 (line 576)
- Rewrite line 623 (title of Table 2)
- In table 2, what is the difference between the two ∆∆G columns?
- Paragraph lines 683-689, it is unclear to me why the NNT-ARM would “relocks the N protein in its closed form”. Why the NT-ARM and not the binding of the RNA alone or the N-N protein contacts between protomers? What happens if the MD simulation is done with a monomer of N bound to a small RNA? What happens if the NC is empty?
- Unknown symbol in the legend of Figure 6 (line 693).
- Paragraph lines 729-745, some of the indicated panels for Figure 7 seem wrong.
- In the legend of Figure S2, remove “showing the region of P68 inserted in the RNA binding groove”.
- In figure S5, panels are not indicated on the figure.
- In figure S6, isn’t there an inversion between panels C and D (F2 vs F1)?
Author Response
We thank the reviewer for the comments and for pointing out some errors, which have been corrected as described here:
1) Since the residues involved in the interaction between P and N have been identified, add a table listing these residues (as supplemental data?).
A table listing the polar interactions has been added as Supplemental Table S1
2) In a paper on parainfluenza virus 5 from Aggarwal and colleagues (Aggarwal et al., 2018, ref. 44 in this paper), the authors used molecular dynamics to look at the opening-closing movement and show that the N protein at the 3’end of the nucleocapsid seems to open more readily than the others. Since Gérard et al. also performed MD simulation on a small nucleocapsid and propose that the absence of the NNT-ARM could favor the opening of N, it would be interesting to compare the movement of the three RABV N protomers of the small nucleocapsid, see the differences in their movements, and compare with PIV5.
A major difference between the MD simulation reported by Aggarwal et al. and that reported here is the time of simulation. They simulated trajectories of 10 ns for each conformation, whereas we simulated several trajectories of about 1 µs for each of the three conformations. We also performed PCA analyses, which provide a better view of the movements over the entire trajectories than a snapshot at a given time
Regarding the analysis of the movement in the three N protomers, the movement of the protomers that are on the 3’ and 5’ not constraint by their neighbors because they are missing the NNT-ARM for one and the NCT-ARM for the other. We thus believe that it is not relevant to interpret the movements in these protomers.
Minor comments:
- In the legend of Figure 1D, specify “red crosses” since one cannot distinguish the shape of the crosses.
This has been corrected
- Idem for Figure 2B (first, put the crosses in red like for Figure 1D).
This has been corrected
- “(F)” is missing in Figure 2.
The label has been added in the Figure
- Homogenize the names of the protein domains (e.g.: PCM and PCM, NT-ARM and NTARM…)
This has been corrected
- Figure 4D, an issue with colors: yellow dots and red line
This has been corrected
- Unknown symbol in the legend of Figure 4 (line 576)
This has been corrected
- Rewrite line 623 (title of Table 2)
The title, legend and table were corrected
- In table 2, what is the difference between the two ∆∆G columns?
See point above. Explanation are now given in the legend of the Table
- Paragraph lines 683-689, it is unclear to me why the NNT-ARM would “relocks the N protein in its closed form”. Why the NT-ARM and not the binding of the RNA alone or the N-N protein contacts between protomers? What happens if the MD simulation is done with a monomer of N bound to a small RNA? What happens if the NC is empty?
We have not performed MD simulations with a single N protomer and a short RNA (9 mer) because in this case there would be neither NNT-ARM nor NNT-ARM as they come from the neighboring subunits, but the reviewer is correct, we have no evidence for this and it was speculation. We have removed this sentence.
- Unknown symbol in the legend of Figure 6 (line 693).
This has been corrected
- Paragraph lines 729-745, some of the indicated panels for Figure 7 seem wrong.
This has been corrected. See also the answers to reviewer 1
- In the legend of Figure S2, remove “showing the region of P68 inserted in the RNA binding groove”.
This has been corrected
- In figure S5, panels are not indicated on the figure.
This has been corrected
- In figure S6, isn’t there an inversion between panels C and D (F2 vs F1)?
This has been corrected